# Chiral phonons in quartz probed by X-rays

Hiroki Ueda[1,2 ✉], Mirian García-Fernández[3], Stefano Agrestini[3], Carl P. Romao[4], Jeroen van den Brink[5,6], Nicola A. Spaldin[4], Ke-Jin Zhou[3] & Urs Staub[1 ✉]

The concept of chirality is of great relevance in nature, from chiral molecules such as sugar to parity transformations in particle physics. In condensed matter physics, recent studies have demonstrated chiral fermions and their relevance in emergent phenomena closely related to topology[1–3]. The experimental verification of chiral phonons (bosons) remains challenging, however, despite their expected strong impact on fundamental physical properties[4–6]. Here we show experimental proof of chiral phonons using resonant inelastic X-ray scattering with circularly polarized X-rays. Using the prototypical chiral material quartz, we demonstrate that circularly polarized X-rays, which are intrinsically chiral, couple to chiral phonons at specific positions in reciprocal space, allowing us to determine the chiral dispersion of the lattice modes. Our experimental proof of chiral phonons demonstrates a new degree of freedom in condensed matter that is both of fundamental importance and opens the door to exploration of new emergent phenomena based on chiral bosons.

Quasiparticles in solids fundamentally govern many physical properties, and their symmetry is of central importance. Chiral quasiparticles are of particular interest. For example, chiral fermions emerge at degenerate nodes in Weyl semimetals[1] and chiral crystals[2,3]. Their chiral characters are directly manifested by a chiral anomaly[7] and lead to enriched topological properties, including selective photoexcitation by circularly polarized light[8], chiral photocurrent[9] and transport[7]. The presence of chiral bosons, such as phonons[4–6,10–17] and magnons[6,18–20], has also extensively been debated.

Chiral phonons are vibrational modes of solids in which the atoms have a rotational motion perpendicular to their propagation with an associated circular polarization and angular momentum. As a result of their angular momentum, chiral phonons can carry orbital magnetic moments, enabling a phonomagnetic effect analogous to the optomagnetic effect from other helical atomic rotations[21,22]. Correspondingly, the phonons can create an effective magnetic field, which has been invoked to explain the observation of excited magnons[23] and enables their excitation through ultrafast angular-momentum transfer from a spin system[24]. Whereas a phononic magnetic field has so far been discussed primarily at the Γ point, chiral phonons naturally arise in noncentrosymmetric materials away from the zone centre and are based on a fundamentally different symmetry.

Experimental observation of phonon chirality has proven to be challenging. If atomic rotations are confined in a plane containing the phonon propagation direction (circular phonons), the mode cannot possess a chiral character (Supplementary Information has symmetry considerations) as occurs for non-propagating phonons at Γ and other high-symmetry points. Therefore, results based on optical-probe techniques, such as chiroptical spectroscopy[16] and circularly polarized Raman scattering[17], are insufficient to identify the presence of chiral phonons because of the large wavelength of optical photons, restricting the exploration very close to the Γ point. The first claim of observation of a chiral phonon was made at the high-symmetry points

of a monolayer transition-metal dichalcogenide[5], although it has been argued to be inconsistent with symmetry arguments[6]. Thus, establishing an experimental method that directly verifies the chiral character of phonons is strongly demanded.

In this work, we demonstrate chiral phonons in a chiral material at general momentum points in the Brillouin zone. We probe the chirality of phonons using resonant inelastic X-ray scattering (RIXS) with circularly polarized X-rays. Our strategy rests on the fact that circularly polarized X-rays are chiral and is inspired by the use of resonant elastic X-ray scattering to probe the chirality of a static lattice by using circularly polarized X-rays on screw-axis forbidden reflections[25]. Using RIXS, circularly polarized chiral photons can couple to dynamic chiral phonon modes by transferring angular momentum, and the process can occur at general momentum points in reciprocal space. Our theoretical analysis shows that the observed circular dichroism in RIXS is caused by the orbitals of the resonant atoms that align in a chiral way determined by the chiral crystal structure; we calculate the angular momentum of the phonons at the corresponding **Q** point using density-functional theory (DFT).

## RIXS considerations

RIXS is a two-step process in which the energy of the incident photon with a given polarization coincides (resonates) with an atomic X-ray absorption edge of the system[26]. For RIXS at the O *K* edge, an incident photon excites an electron from the O 1*s* inner shell to the 2*p* outer shell. The combined core hole and excited electron form a short-lived excitation in this intermediate state that interacts with the lattice and creates phonons as it deforms its local environment[27,28]. The final RIXS step involves the deexcitation of the electron from 2*p* to 1*s*, causing the emission of a photon while leaving behind a certain number of phonons in the system. The detected energy and momentum of the emitted photon are directly related to the energy and momentum of the phonon created in the solid.

[1]Swiss Light Source, Paul Scherrer Institute, Villigen, Switzerland. [2]SwissFEL, Paul Scherrer Institute, Villigen, Switzerland. [3]Diamond Light Source, Didcot, UK. [4]Department of Materials, ETH Zurich, Zurich, Switzerland. [5]Institute for Theoretical Solid State Physics, IFW Dresden, Dresden, Germany. [6]Institute for Theoretical Physics and Würzburg–Dresden Cluster of Excellence ct.qmat, Dresden University of Technology, Dresden, Germany. ✉e-mail: hiroki.ueda@psi.ch; urs.staub@psi.ch

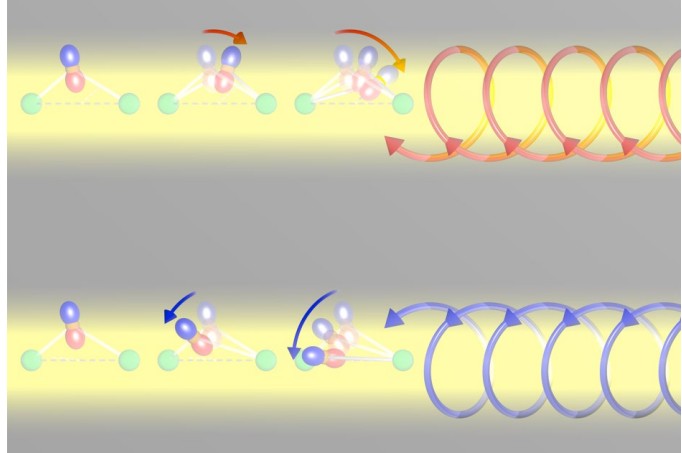

**Fig. 1 | Angular-momentum transfer in an RIXS experiment.** The angular momentum of the photons (opposite between C+ (up, red) and C− (down, blue)) is transferred to a crystal, causing a revolution in this case of anions (orange spheres with *p* orbitals) relative to their neighbouring cations (green spheres).

To illustrate the mechanism by which RIXS excites chiral phonons in quartz, we consider an Si–O chain in which the O ions bond to the Si ions via the 2*p* orbital pointing toward the central axis of the chain (Fig. 1 and Supplementary Fig. 1). While this O 2*p* orbital is unchanged in the local frame of the ligand as it revolves around the central axis with angle $\phi$, in the global frame its direction changes upon revolution. We describe the spatial coordinate of the phonon by the angle $\phi$ and denote the creation operator of an electron in the 2*p* orbital along the global *x* axis as $p_x^\dagger$ and along the global *y* axis as $p_y^\dagger$. We construct the RIXS intermediate-state Hamiltonian $H_I$ such that during an (adiabatically slow) revolution of the atom around the *z* axis, the ground-state wave function always points toward the centre of rotation (Supplementary Information has a detailed derivation):

$$H_I = -\alpha ss^\dagger \mathbf{p}^\dagger (\sigma_z \cos 2\varphi + \sigma_x \sin 2\varphi)\,\mathbf{p}, \tag{1}$$

where $ss^\dagger$ is the core-hole density operator, the vector operator $\mathbf{p} = (p_x, p_y)$, and $\sigma_i$ denotes the Pauli matrixes with $i = x, y, z$. The RIXS operator that takes the system from the ground state |0> to the final state |*f*> with *m* phonon modes can be evaluated to lowest order in $\alpha$ using the ultrashort core-hole lifetime expansion[27]. Introducing the circular polarization basis $\boldsymbol{\epsilon}_c$, where a fully left circularly polarized photon corresponds to $\boldsymbol{\epsilon}_c^L = (1,0)$ and a right one to $\boldsymbol{\epsilon}_c^R = (0,1)$, the RIXS amplitude becomes (Supplementary Information)

$$A_m = (\boldsymbol{\epsilon}_c')^* \langle m | \sigma_z e^{2i\varphi\sigma_y} | 0 \rangle \boldsymbol{\epsilon}_c. \tag{2}$$

This shows that angular momentum is transferred to the phononic system when the incident and scattered photons have different circular polarization. Figure 1 shows conceptually how such interactions between circularly polarized photons and the lattice can launch revolutional lattice vibrations through this angular-momentum transfer.

## Experiments

As our target material, we choose the prototypical chiral crystal quartz ($\alpha$-SiO$_2$), in which SiO$_4$ tetrahedra form a chiral helix along [001] (Fig. 2). The resulting chiral space group is either $P3_221$ (left quartz) (Fig. 2a) or $P3_121$ (right quartz) (Fig. 2b). A recent DFT study[15] pointed out the chirality and phonon angular momentum of some phonon branches and demonstrated the reversal of chirality between opposite enantiomers, as well as the absence of phonon angular momentum at the Γ point.

We performed RIXS experiments with circular polarization (C+/C−) on two quartz crystals with opposite chirality. With incident photon energy tuned around the O *K* edge reaching an energy resolution of approximately 28 meV, we collected spectra at $\mathbf{Q}_1 = (-0.25, 0, 0.32)$ reciprocal lattice units (Fig. 2c; Methods has details). The spectrum for various incident photon energies (shown in Fig. 3) shows clear peaks on the energy-loss side at resonance, which become suppressed for energies farther away from resonance. Note that the energy resolution is insufficient to assign the peaks to individual phonons[29]. All peaks above the energy of the highest phonon mode of approximately 0.2 eV (ref. 29) are the result of higher-harmonic phonon excitations.

Figure 4 shows the C+ and C− RIXS spectra from left-handed (Fig. 4a) and right-handed (Fig. 4b) quartz and their dichroic contrasts (Fig. 4c) at 20 K. We see a clear contrast between C+ and C−, and the dichroism changes sign for the opposite chiral enantiomers, indicating that it is caused by chirality of the modes. We find similar contrast between C+ and C− at the other reciprocal points with different RIXS spectra due to different phonon energies (dispersion) and different RIXS cross-sections (Supplementary Fig. 2). These observations demonstrate unambiguously that circularly polarized photons couple to chiral phonons, with the chirality of the phonons defined by the lattice chirality, and that RIXS with circularly polarized X-rays can be used to probe phonon chirality.

## DFT and discussion

We use DFT to calculate the phonon dispersion and phonon circular polarization for all phonon branches and show their dispersion between $\mathbf{Q}_1$ and Γ in Fig. 5a for right quartz (details are found in Methods; Supplementary Figs. 4 and 5 show other directions in reciprocal spaces and components of the circular polarization vector). Note that, since we are interested in low symmetry points in the Brillouin zone, we show a different direction from that in ref. 15, as well as additional bands. The colour scale indicates the phonon circular polarization (**S**) (ref. 4),

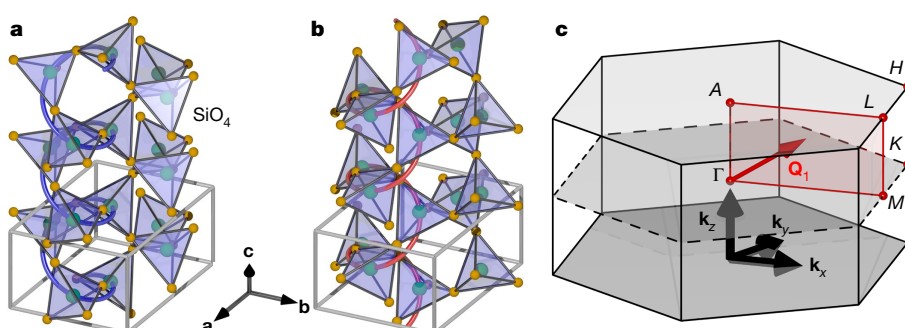

**Fig. 2 | Crystal structure and Brillouin zone of quartz. a–c**, Crystal structures of left quartz (**a**) and right quartz (**b**) and the Brillouin zone with $\mathbf{Q}_1$, where the RIXS spectra have been taken (**c**).

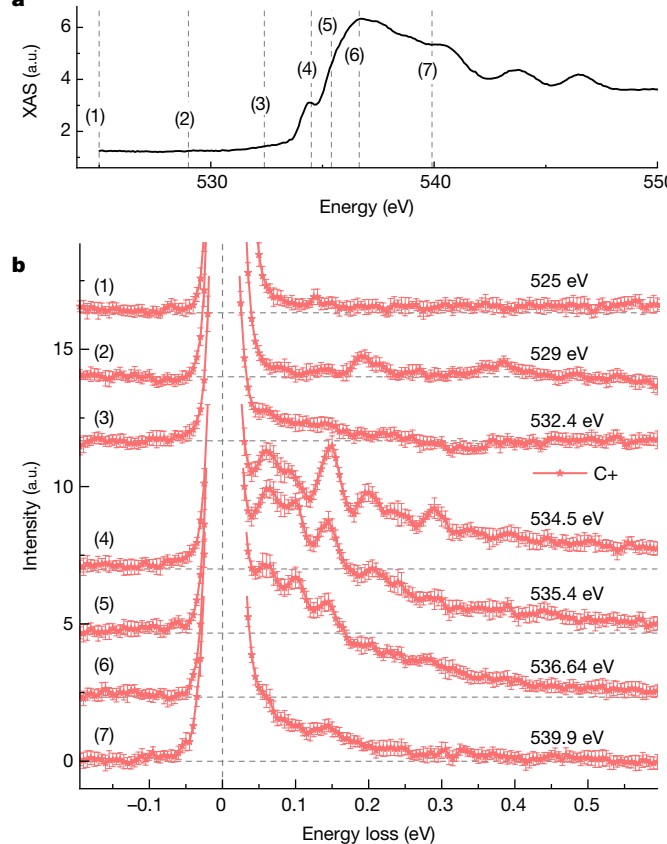

**Fig. 3 | XAS and photon-energy dependence of RIXS. a**, X-ray absorption spectrum around the O $K$ edge. **b**, RIXS spectra taken with C+ for left-handed quartz at $\mathbf{Q}_1 = (-0.25, 0, 0.32)$ for the incident photon energies indicated by the dashed lines in **a**. Each spectrum in **b** is vertically shifted to enhance visibility. Error bars are smaller than the line width in **a** and in standard deviation in **b** (Methods).

which indicates the chirality of a phonon mode; it is defined, for example, for the $z$ component $S_z$ as

$$S_z = \sum_{m=1}^{n} S_{z,m} = \sum_{m=1}^{n} (|\langle r_{m,z} | \epsilon_m \rangle|^2 - |\langle \ell_{m,z} | \epsilon_m \rangle|^2). \quad (3)$$

Here, $\epsilon_m$ are the phonon eigenvectors of each of the $n$ atoms in the unit cell (normalized such that $\sum_m |\langle \epsilon_m | \epsilon_m \rangle| = 1$), and $|r_{m,z}\rangle$ and $|\ell_{m,z}\rangle$ are eigenvectors corresponding to pure right- and left-handed rotations. The phonon angular momentum (**L**) is then given by $\mathbf{L} = \hbar \mathbf{S}$ (ref. 4). We also report the mode effective charges (Fig. 5b) as a metric of the strength of the interaction between the mode and light, calculated following the method of ref. 30.

When we match the calculated and measured modes, we find that those with strong dichroic contrast are those calculated to have a large chirality. The peak with the largest contrast is at approximately 50 meV for all the reciprocal points we measured ($\mathbf{Q}_1$ in Fig. 4c and $\mathbf{Q}_2 = (-0.29, 0.14, 0.32)$ and $\mathbf{Q}_3 = (-0.25, 0.25, 0.32)$ in Supplementary Fig. 2), suggesting that a mode that has large phonon circular polarization and energy around 50 meV dominates the contrast. The mode at the energy of approximately 47.6 meV at $\mathbf{Q}_1$, which we refer to as mode X, matches the conditions (Supplementary Fig. 4 and Supplementary Table 1, which tabulates the energy and phonon circular polarization of all phonon modes at the measured **Q** points). Figure 5c and Supplementary Video 1 visualize mode X at $\mathbf{Q}_1$ and show that it involves a circular motion of the atoms. Importantly, the mode satisfies the symmetry requirement for a chiral phonon mode.

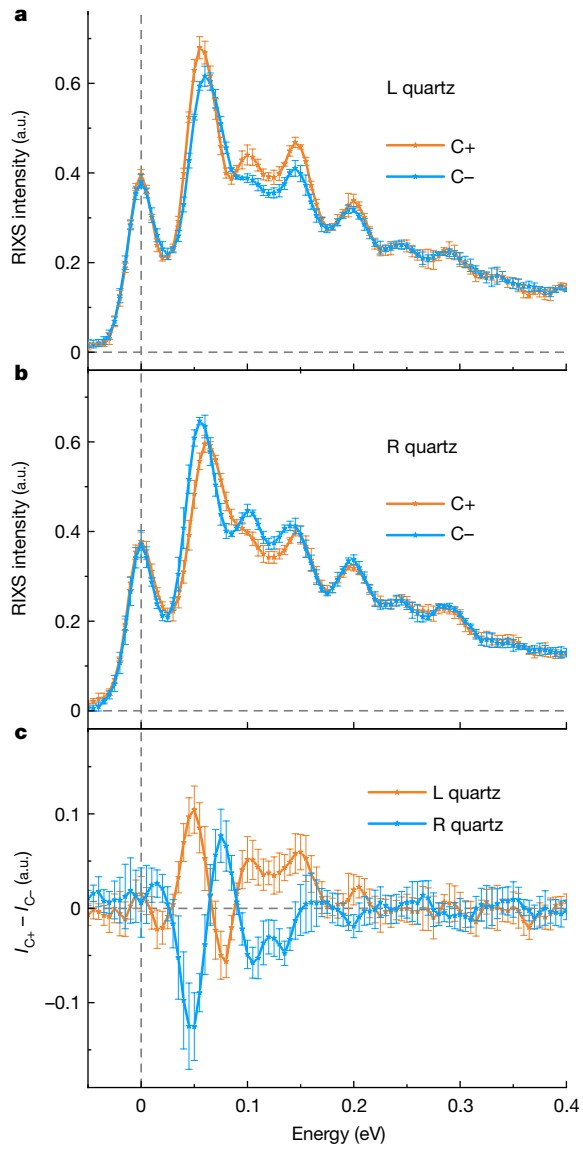

**Fig. 4 | RIXS with circularly polarized X-rays. a,b**, Comparison between left (L) quartz (**a**) and right (R) quartz (**b**) taken at the incident photon energy of 534 eV and $\mathbf{Q}_1 = (-0.25, 0, 0.32)$. **c**, Extracted circular dichroic components of the data shown in **a** and **b**. Error bars are in standard deviations.

For non-magnetic quartz, the RIXS spectra at the O $K$ edge are mainly sensitive to the O 2$p$ orbital states. This means that phonon modes that significantly affect, for example, the orientation of the 2$p$ orbital states will create large scattering contrast in RIXS and will also be strongly X-ray polarization dependent. Figure 5d and Supplementary Video 2 visualize the evolution of the local charge quadrupoles at the O site when the chiral phonon mode is excited (Fig. 5c or Supplementary Video 1). These charge quadrupoles reflect the time evolution of the O 2$p$ orbitals, which shows that the dichroic RIXS signal is due to an evolution of the chiral stacking of the O 2$p$ orbital moments in the chiral phonon excitation as described in equations (1) and (2).

Note that the mode with the largest contrast is not the phonon mode with the largest phonon circular polarization. Instead, the mode has a large mode effective charge at $\mathbf{Q}_1$, as shown in Fig. 5b. This indicates that the contrast depends on not only the chiral amplitude of a mode itself but also, the modulation of the electronic charges with respect to the plane of the electric fields of the circularly polarized X-rays. Note that there is an additional consideration. Phonon circular polarization specifies a preferred revolution direction of atoms in the excitation,

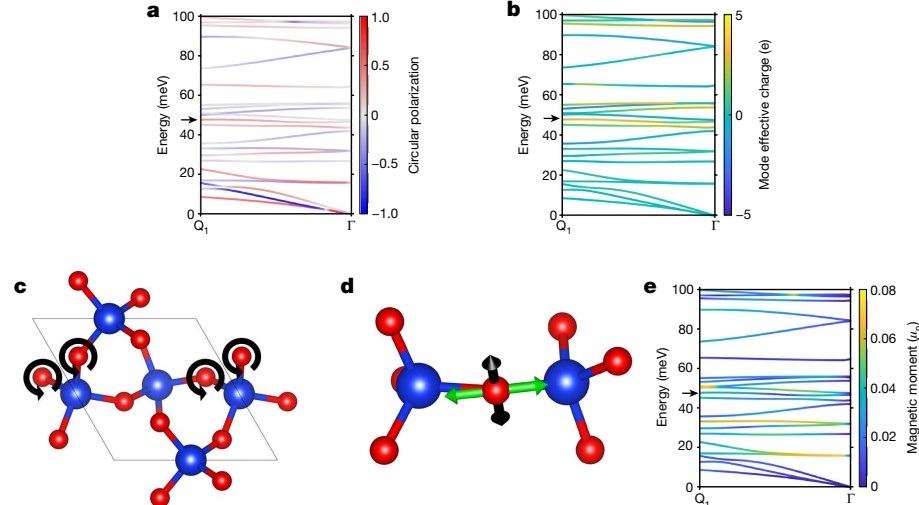

**Fig. 5 | Phonon dispersion and chiral phonon mode. a**, Low-energy phonon dispersion for right quartz along the Γ to $Q_1$ direction. Colours represent the *z* component of the phonon circular polarization. **b**, The same phonon band structure with colours representing mode effective charges (a measure of the degree in which the electronic charge distribution is perturbed by the phonons) in units of the elementary charge. **c**, The chiral phonon mode at $Q_1 = (-0.25, 0, 0.32)$ (indicated with an arrow in **a**) showing the main chiral revolutions of the oxygen atoms that have a different phase along the chain.

**d**, Associated change in the local quadrupole moment (associated with the O 2*p* orbital) for a revolving oxygen atom between the phonon at phase 0 and phase π (black vectors represent an increase in the atomic quadrupole moment between its position at phonon phase 0 and its position at phase π, and green vectors represent a decrease). **e**, The phonon band structure coloured according to the magnitude of the magnetic moment of the phonons in units of the nuclear magneton.

which can only be excited with the matching circular photon polarization (Fig. 1). As modes of opposite chirality have different energies (in Supplementary Fig. 4 and Supplementary Table 1, modes with opposite chirality, degenerated at the Γ point, split at away from the zone centre), the peaks that are composed of several modes show a peak shift when taken with opposite circular polarization (Fig. 4).

In Fig. 5e, we show the associated magnetic moments induced by the chiral motion of the charged ions in the chiral phonons, which we calculate by extending the method used in refs. 21,22 so that it is applicable at an arbitrary point in **Q** space. We begin by constructing the atomic circular polarization vector $S_m$ as $S_m = [S_{x,m} S_{y,m} S_{z,m}]$ (equation (3)), yielding the angular momentum of each atom as $L_m = \hbar S_m$. The magnetic moment ($\mu_m$) of each atom participating in the phonon is

$$\mu_m = L_m \gamma_m = \hbar S_m Z_m / 2m_m, \qquad (4)$$

where $\gamma_m$ is the gyromagnetic ratio tensor, which is derived from $Z_m$, the Born effective charge tensor, and $m_m$, the atomic masses. The phonon magnetic moment is then simply $\mu = \sum_{m=1}^{n} \mu_m$. We show our calculated mode- and **Q**-point resolved magnetic moments in Fig. 5e and see that chiral phonons in quartz carry magnetic moments throughout the Brillouin zone, although the calculated magnetic moments are relatively small due to the low values of $\gamma_m$. These phonon magnetic moments do not normally create a net magnetization due to the presence of time-reversal related pairs with opposite chirality and magnetic moment. If time-reversal symmetry is broken, however, population imbalances between the chiral pairs can be created[31]. Figure 5e also suggests that the phonon chirality can be investigated directly through interactions with the phonon magnetic moment using, for example, polarized inelastic neutron scattering.

In conclusion, we have used RIXS with circularly polarized X-rays to demonstrate the chiral nature of the phonons in chiral quartz crystals and in turn, have established a fundamental methodology for characterizing chiral phonons. With the technique established by this proof-of-principle study, the chirality of phonons at general momentum points can be characterized, opening up new perspectives in chiral

phononics. For example, our work indicates that RIXS can be used to quantify the role of chiral phonons in exotic phenomena proposed in topological materials[10,32–34], as well as to characterize interactions such as electron and spin couplings with chiral phonons[14,35–38].

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

# Article

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

# Methods

## RIXS

RIXS measurements were performed at Beamline I21 at the Diamond Light Source in the UK[39]. Used photon energy is around the O $K$ edge, and polarization is circular (C+/C−). The energy resolution is estimated as 28 meV from the full-width of the half-maximum of the elastic peak from a carbon tape. Enantiopure single crystals purchased commercially have the widest face perpendicular to the [001] axis. The manipulator installed at the beamline allows us to rotate the crystal along the azimuthal angle, enabling us to access different momentum points during the experiment: $Q_1 = (−0.25, 0, 0.32)$, $Q_2 = (−0.29, 0.14, 0.32)$ and $Q_3 = (−0.25, 0.25, 0.32)$. We defined error bars in an RIXS spectrum as the standard deviation of individual scans from their average spectrum. X-ray absorption spectroscopy obtained before the RIXS measurements is based on the total electron yield method.

## DFT

Density-functional calculations were performed using the Abinit software package (v.9)[40,41] and the Perdew–Burke–Ernzerhof exchange–correlation functional[42] with the dispersion correction of ref. 43. The phonon band structure was determined using density-functional perturbation theory[40] using norm-conserving pseudopotentials, a 38-Ha plane wave energy cutoff, an $8 × 8 × 8$ Monkhorst–Pack grid in $k$ space[44] and a $4 × 4 × 4$ grid in $Q$ space. Calculations of the electronic and phononic structure were additionally performed explicitly at the experimentally measured $Q$ points. Frozen phonon calculations were performed using the projector-augmented wave method[45] to obtain local quadrupole moments with the multipyles postprocessing script[46]. These calculations used a 192-Ha plane wave energy cutoff within the atomic spheres and a 32-Ha cutoff without. The default pseudopotentials and projector-augmented wave datasets from the Abinit library were used.

## Data availability

Experimental and model data are accessible from the Paul Scherrer Institute Public Data Repository[47]. Source data are provided with this paper.

## Code availability

The MATLAB code used to obtain phonon circular polarizations and magnetic moments from Abinit output is available at https://github.com/cpromao/phonon_polarization.

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

**Acknowledgements** We thank A. Nag for advising on data analysis and stimulating discussion. The resonant inelastic X-ray scattering experiments were performed at beamline I21 at the Diamond Light Source (proposal MM28375). H.U. was supported by the National Centers of Competence in Research in Molecular Ultrafast Science and Technology (grant 51NF40-183615) from the Swiss National Science Foundation and from the European Union's Horizon 2020 Research and Innovation programme (Marie Skłodowska-Curie Grant 801459–FP-RESOMUS). This work was funded by the European Research Council under the European Union's Horizon 2020 Research and Innovation programme (grant 810451). Computational resources were provided by ETH Zurich and the Swiss National Supercomputing Centre (project eth3). C.P.R. acknowledges the support of the European Union and Horizon 2020 through the Marie Skłodowska-Curie Fellowship (grant no. 101030352). J.v.d.B. thanks the Deutsche Forschungsgemeinschaft for support through the Würzburg–Dresden Cluster of Excellence on Complexity and Topology in Quantum Matter ct.qmat (EXC 2147 Project no. 39085490) and the Collaborative Research Center SFB 1143 (project no. 247310070).

**Author contributions** H.U. and U.S. conceived and designed the project. H.U., M.G.F., S.A., K-J.Z. and U.S. performed resonant inelastic X-ray scattering experiments. H.U. analysed the experimental data. C.P.R. and N.A.S. performed density-functional theory calculations. J.v.d.B. contributed to the mechanism by which resonant inelastic X-ray scattering excites chiral phonons. H.U., C.P.R., J.v.d.B., N.A.S. and U.S. wrote the manuscript with contributions from all authors.

**Funding** Open Access funding provided by Lib4RI – Library for the Research Institutes within the ETH Domain: Eawag, Empa, PSI & WSL.

**Competing interests** The authors declare no competing interests.

**Additional information**
**Correspondence and requests for materials** should be addressed to Hiroki Ueda or Urs Staub.
