## [Peer Review File · Nature]

Manuscript Title: Chiral phonons in quartz probed by X rays

Reviewer Comments & Author Rebuttals

Reviewer Reports on the Initial Version:

Referees' comments:

Referee #1 (Remarks to the Author):

The authors present a thorough experimental and theoretical work introducing a new resonant inelastic x-ray experimental technique to probe chiral phonons in materials. This involves utilizing circularly polarized x-rays, which carry angular momentum, to directly probe chiral phonon excitations in both energy and, owing to the large momentum available in x-ray photons, momentum space. Such a technique is sorely needed given the current emphasis and progress on topological phenomena in condensed matter systems. Chirality being a key feature of this new paradigm of phases which do not conform with Landau theory of phase transitions. The experimental evidence presented on the prototypical chiral quartz system is compelling and clearly supports their conclusions. Furthermore, the theoretical backing confirms the experimental findings and provides a blueprint for future work on chiral phonons in the more interesting quantum materials this technique will be heavily utilized on. To the best of my knowledge, this work is the first conclusive evidence of the ability to directly probe chiral excitations with resonant inelastic x-ray scattering. It is very likely this work will lead to immediate and rich results on chiral phonons and indeed other chiral excitations using the well-established RIXS spectrometers available at numerous synchrotrons throughout the world. Data presented is compelling and figures are of high quality making them appropriate for publication. Error bars and analysis are present and appear reasonable. The writing is also clear and concise.

In conclusion, the authors present a very convincing study introducing a new resonant inelastic x-ray scattering technique for probing chiral excitations which has a very high likelihood of leading to exciting findings in a very topical field of current study. The manuscript in its current form is suitable for publication in Nature.

Derek Meyers

Referee #2 (Remarks to the Author):

Key results:

Chirality is a symmetry with high a relevance in all natural sciences. The chirality of phonons, i.e. quantised excitations of the crystal lattice, is of special interest in condensed-matter physics, since it is related to the angular momentum of phonons, a quantity the investigation of has only just begun.

Using resonant inelastic X-ray scattering with circularly polarised X-rays (RIXS) the authors demonstrate the chiral nature of phonons in a chiral quartz crystal.

Data & methodology:

The study was carefully conducted, RIXS measurements are supported by symmetry considerations and DFT calculations, a very good balance of theory and experiment. I have no doubts regarding the validity of the data and conclusions drawn from the data.

Clarity of paper:

The paper is well written and understandable, previous literature is well cited, and the abstract, the introduction and conclusions are appropriate.

I have, however, the impression that the quality of the figures could be improved, especially Figs. 4 and 5. Here the labels are very small and in Fig 5a the lines can hardly be seen (at least in my copy of the manuscript).

I would also recommend to improve the figure captions. In the caption of Fig 3 it says "Error-bars are smaller as the line". Probably, the authors meant "Error bars are smaller than the line thickness". Is that true for Fig. 3a?

Furthermore, in the caption for Fig 3b) it is probably meant "Error bars show the standard deviation (SD)". Here, I could not find any explanation how the SD is calculated in the text. Similar criticism applies to the caption of Fig. 4.

Finally, I have one further remark. I noted that, when describing the atomic motion, the authors - only once - use the term "revolution" instead of "rotation" (caption of Fig.5). In fact, revolution - even though less common - is probably even the better term, since the atoms move on a circular path without spinning (as also assumed and shown for their p-orbitals in Fig. S1). This is an interesting detail the authors might consider to mention somewhere.

Originality and significance:

The concept of chiral phonons has already been known from theoretical work in 2015 (Ref.4). Then, in 2018 (Ref. [5]) the first experimental observation of chiral phonons was published in Science. In Ref. 6, however, it is claimed that this finding is a misinterpretation since in that experiment, the motion of the nuclei is in the sample plane (a monolayer transition-metal dichalcogenide) that must also contain the k-vector, and, hence, these phonons would not be chiral. Considering this, the submitted work might well be the first direct experimental proof of the chirality of phonons in a chiral crystal.

Conclusion:

The understanding of chirality is of general importance in all natural sciences. The direct observation of this chiral quasi-particle is a significant finding and of immediate interest to many people in condensed-matter physics. Chiral phonons carry angular momentum and due to spin phonon coupling - in connection with angular momentum conservation in the total system - the angular momentum of phonons should couple to the magnetisation of a magnetic material. The

understanding of chiral phonons will, hence, in the future also contribute to the understanding and exploitation of spintronic phenomena. Over all, I think that these findings are sufficiently significant to warrant publication of the submitted paper in Nature, after my remarks above have been considered.

Referee #3 (Remarks to the Author):

This manuscript presents an observation of chiral phonons at finite momentum in chiral quartz crystals using resonant inelastic x-ray scattering (RIXS) with circularly polarized x-ray. The experimental results agree well with DFT calculations of chiral phonon modes in the material. Chiral phonons are of great importance in materials science because they exhibit unique and useful properties that can be exploited in a range of applications. The capability of RIXS to determine chiral phonons may interest a broad readership. The manuscript is well-written and clear. However, the impact and generality of this observation and method require clarification.

One point that needs clarification is the requirement for the observation of chiral phonons by RIXS. It would be helpful to know if breaking a particular crystal symmetry, such as spatial inversion, mirror reflection, and roto-inversion, or a combination of the three is necessary. The authors show that the circular dichroic signal exhibits the opposite sign for left-handed and right-handed quartz. If a material only has one-handedness, either left-handed or right-handed, does a circular dichroic in RIXS spectra necessarily lead to the conclusion of chiral phonon? Is there any other symmetry breaking that can also explain such circular dichroism?

Additionally, can this method be widely applicable to look for chiral phonons if the crystal chirality is unknown, and help to classify chiral crystals? Or can it only demonstrate the presence of chiral phonons in well-established chiral materials? The current work seems to address the latter and requires prior knowledge of the crystal structure and symmetry of the material firsthand. If the chiral phonon is already expected in a chiral crystal structure, then only confirming the chirality experimentally may not qualify for publication in Nature, but more suitable for a specialized journal.

Regarding the method, the measurement is performed at O K-edge due to O ions bond vibrations, which requires an ultrashort core-hole lifetime in the intermediate state for the RIXS process. As different absorption edges have different intermediate lifetimes and are sensitive to different excitations, it would be helpful to know if this method can be used for broad materials without oxygen. Is this method limited to materials with oxygen since only the phonons are excited at O K-edge? It seems things will become complicated at other edges since RIXS spectra can probe various collective excitations at one time, such as spin, orbital, and charge excitations, besides phonons. It would be difficult to separate chiral phonons in the presence of so many excitations.

Finally, besides identifying the phonons as chiral, it would be interesting to know if there is more quantitative information, such as coupling strength, that can be extracted from this kind of RIXS measurements with circularly polarized X-rays.

Error bars are missing in the data. The authors should add error bars to the RIXS spectra in Figures 3 and 4 to demonstrate intrinsic dichroic signals over noise levels. Additionally, the authors should add the coordinate axis (K_x , K_y , K_z) to the Brillouin zone in Fig. 2c, as it is hard to read the Q values without the axis.

Author Rebuttals to Initial Comments:

Dear Tobias

Thank you for providing us the referee generally positive reports and the asking for revisions and answers to the raised questions on the implications/impact of our work, which we give below.

Reply to the referee report

Let us first thank the referees for the careful reading and the constructive and positive evaluation of our paper.

Below we give a point-to-point answer to the raised points. Referee comments are given in blue, our reply in black and changes made to the manuscript are in red.

Referee 1:

We very much appreciate the positive evaluation of this referee, who judges our paper as publishable as is.

Referee 2:

Again, very much appreciate the positive evaluation of this referee, judging that paper to be published after the consideration of a few remarks.

- 1) "I have, however, the impression that the quality of the figures could be improved, especially Figs. 4 and 5. Here the labels are very small and in Fig 5a the lines can hardly be seen (at least in my copy of the manuscript). "

We enlarged the labels of the figures.

- 2) I would also recommend to improve the figure captions. In the caption of Fig 3 it says "Error-bars are smaller as the line". Probably, the authors meant "Error bars are smaller than the line thickness". Is that true for Fig. 3a?

Thank you for pointing out our unprecise wording. Errors are smaller than the thickness of the lines in Figure 3a.

We changed the caption to

“Errorbars are smaller than the line **width** in (a) and in Standard Deviation (SD) **see Methods (b).**”

- 3) Furthermore, in the caption for Fig 3b) it is probably meant "Error bars show the standard deviation (SD)". Here, I could not find any explanation how the SD is calculated in the text. Similar criticism applies to the caption of Fig. 4.

The referee is correct. The following sentence has been added to the Method section describing how error bars have been obtained.

We defined error bars in a RIXS spectrum as the standard deviation of individual scans from their average spectrum.

- 4) Finally, I have one further remark. I noted that, when describing the atomic motion, the authors - only once - use the term “revolution” instead of “rotation” (caption of Fig.5). In fact, revolution - even though less common - is probably even the better term, since the atoms move on a circular path without spinning (as also assumed and shown for their p-orbitals in Fig. S1). This is an interesting detail the authors might consider to mention somewhere.

The point of revolution versus rotation is an excellent point; indeed the atoms in our chiral phonons are revolving / orbiting around a central point rather than rotating around their own axes. We replaced several **rotations** with **revolutions** in the manuscript.

Referee 3

The referee points out the importance of chiral phonons and that the capabilities of detecting them are of broad interest, but he/she likes to have some clarifications on the method used.

- 1) One point that needs clarification is the requirement for the observation of chiral phonons by RIXS. It would be helpful to know if breaking a particular crystal symmetry, such as spatial inversion, mirror reflection, and roto-inversion, or a combination of the three is necessary. The authors show that the circular dichroic signal exhibits the opposite sign for left-handed and right-handed quartz. If a material only has one-handedness, either left-handed or right-handed, does a

circular dichroic in RIXS spectra necessarily lead to the conclusion of chiral phonon? Is there any other symmetry breaking that can also explain such circular dichroism?

The referee raises two distinct points in this comment. Regarding the symmetry breaking, the distinction between a *circularly polarized* phonon, which possesses an associated angular momentum, and a *chiral* phonon, which also contains a direction of motion (propagation) that breaks all mirror symmetries, is relevant here. As the origin of the observed circular dichroism in RIXS is a change of circular polarization of the incident photon and a transfer of angular momentum to the sample, we would expect that, in addition to the chiral phonons that cause the effect described here, *nondegenerate* circularly polarized phonons should also show circular dichroism. Note, that this still requires experimental confirmation! Therefore, the symmetry requirement (for a diamagnetic material) to observe circular dichroism in RIXS is the absence of inversion symmetry, as nondegenerate circularly polarized phonons exist in noncentrosymmetric (both chiral and polar) materials away from the high-symmetry points of the Brillouin zone. In addition, we would like to emphasize the distinction between the chiral modes that we identify here and the trivial case of circularly polarized phonon modes that are formed from a linear combination of *degenerate* linearly polarized modes. While the latter can carry angular momentum and couple to magnetism, displaying for example the phonon Zeeman effect [PR Materials 1 014401 (2017)], the modes of opposite angular momentum are not split in energy and so they do not show circular dichroism.

The other point raised here is whether there is any other non-phononic mechanism that creates circular dichroism in RIXS. Indeed, there is, the most straightforward being chiral magnetic excitation that disperses in magnetic materials as very recently observed in [ACS Appl. Mater. Interfaces 2019, 11, 36213–36220]. (see also reply to point 3 below). As quartz does not contain any unpaired spins, this is of no relevance here.

- 2) Additionally, can this method be widely applicable to look for chiral phonons if the crystal chirality is unknown, and help to classify chiral crystals? Or can it only demonstrate the presence of chiral phonons in well-established chiral materials? The current work seems to address the latter and requires prior knowledge of the crystal structure and symmetry of the material firsthand. If the chiral phonon is already expected in a chiral crystal structure, then only confirming the chirality experimentally may not qualify for publication in Nature, but more suitable for a specialized journal.

The existence and behavior of chiral phonons and their influence on material properties are of great current research interest, as outlined in the introduction and confirmed by the other two referees. Therefore, it is essential to have a method that can demonstrate the chirality of phonons when the crystal structure is not easily determined, as e.g. in thin films or van der Waals materials. This was lacking so far. In this work we demonstrate that RIXS has the power to investigate phonons in

materials at different points in reciprocal space and so can confirm the absence of a centre of symmetry. This concept can be readily extended to materials in which the symmetries can only be addressed indirectly.

While the circular dichroism of a phonon cannot directly be used to determine the enantiomeric identity of a crystal *a priori*, this could be possible in combination with theoretical modeling techniques such as density functional theory (DFT). For example, in the current work we calculate explicitly using DFT the direction of the phonon splitting in each enantiomer, allowing us to associate each phonon excitation with the corresponding enantiomer.

- 3) Regarding the method, the measurement is performed at O K-edge due to O ions bond vibrations, which requires an ultrashort core-hole lifetime in the intermediate state for the RIXS process. As different absorption edges have different intermediate lifetimes and are sensitive to different excitations, it would be helpful to know if this method can be used for broad materials without oxygen. Is this method limited to materials with oxygen since only the phonons are excited at O K-edge? It seems things will become complicated at other edges since RIXS spectra can probe various collective excitations at one time, such as spin, orbital, and charge excitations, besides phonons. It would be difficult to separate chiral phonons in the presence of so many excitations.

Again, this point contains several aspects. 1) The core-hole life time is not relevant for the RIXS experiment as the incident energy is fixed. It is although of importance when varying the incident energy. 2) The method is applicable in principle for any absorption edge of atoms in a material. Of course, there are technical issues, such as energy resolution and cross sections, which might make an experimental observation of chiral phonons challenging. 3) We do not see any complications with orbital excitations, as these excitations lie in a completely different energy regime (eV's). 4) In a "non-magnetic" material, there is of course no entanglement with magnetic excitations, although we note that the finding here is that a chiral mode carries a "magnetic moment" as explained in the text. Most magnetic materials contain, in addition to the magnetic ions, additional ions, for example oxygen ligands, that do not carry a spin. Hence, by choosing an X-ray energy that probes such an absorption edge, RIXS is not directly sensitive to magnetic excitations. If choosing a magnetic ion absorption edge, then indeed the situation will be more complex. In this case, the excitation energies might still be sufficiently different to be separated, and / or to reveal additional physics related to new couplings.

- 4) Finally, besides identifying the phonons as chiral, it would be interesting to know if there is more quantitative information, such as coupling strength, that can be extracted from this kind of RIXS measurements with circularly polarized X-rays. With the combination of RIXS and DFT we are able to obtain quantitative information about the interactions. For example, the coupling strength between the atoms, which determines the dispersion of the modes, can be directly extracted from our DFT calculations, and benchmarked by confirming that our DFT mode

frequencies match the measured values. Without the combined theory / experiment approach, extraction of quantitative information will certainly be more challenging.

- 5) Error bars are missing in the data. The authors should add error bars to the RIXS spectra in Figures 3 and 4 to demonstrate intrinsic dichroic signals over noise levels. Additionally, the authors should add the coordinate axis (K_x , K_y , K_z) to the Brillouin zone in Fig. 2c, as it is hard to read the Q values without the axis.

We thank the referee for pointing this out. Errorbars have been added where missing. Please see also the reply to point 3 from referee 2. We also added the suggested labels to Fig. 2 and hope this gets the figure more clear now.

Changes made to the manuscript:

In addition to the very few text changes as pointed out above in red, we updated figures 2, 4 and 5.

Reviewer Reports on the First Revision:

Referees' comments:

Referee #3:

Recommends publication